# Obtention and preliminary clinical evaluation of an equine albumin for intravenous administration in adult Colombian Creole Horses (*Equus ferus caballus*)

Vanessa Cifuentes[1,2*], Angélica Maria Zuluaga-Cabrera[2], Leidy Johana Vargas-Muñoz[2], Sebastián Estrada-Gómez[1,2*]

**1** Grupo de Toxinologia y Alternativas Terapeuticas Alimentarias, Universidad de Antioquia UdeA, Medellin, Antioquia, Colombia, **2** Tech Life Saving (TLS), a tech innovation group company, Medellin, Antioquia, Colombia

* sebastian.estrada@udea.edu.co (SEG); vanessa.cifuentes@udea.edu.co (VC)

## Abstract

Albumin is one of the most abundant and physiologically important blood protein in horses due to its ability to regulate vascular volume and transport blood metabolites or drugs. Despite the importance of this protein, in Colombia there is no previous reference of the use of equine albumin in horses as a pharmacological therapy and there is no available any pharmaceutical preparation of this protein to be administrated in horses. This study aims to evaluate for first time the preliminary clinical response of healthy adult Colombian Creole horses after the intravenous administration of an equine albumin preparation, manufactured in Colombia. Equine albumin was prepared from the plasma of healthy horses and obtained through the modified Salting Out technique. The Standard Quality Characterization was carried out following World Health Organization standards which included physicochemical, sterility and hemotropics tests before being administered to the horses. Albumin was administered at a concentration of 5,334 mg per animal to 3 healthy horses that were clinically evaluated before, during and after albumin administration, recording different paraclinical and clinical parameters. After manufacturing, the equine albumin obtained fulfilled the quality characteristics to be administered intravenously. After the administration, the product did not generate any adverse reactions or adverse clinical alteration at the concentration used. During the clinical evaluation we were able to observe a plasma volume expansion. Results indicates the ability to obtain a high quality product that can potentially be used as a pharmacological therapy in horses.

**Data availability statement:** All relevant data are within the manuscript and its Supporting Information files.

**Funding:** This research was funded by Tech Life Saving and the Universidad de Antioquia through its program CODI with the grant number ES84200134. Sebastian Estrada-Gomez is the recipient of the related grants.

**Competing interests:** The authors have declared that no competing interests exist. Tech Life Saving (TLS) is a Spin-Off from the University of Antioquia where different professors and researchers from the University of Antioquia participate. TLS is the owner (by licensing of the University of Antioquia) of the manufacturing and quality control process and is covered by an intellectual property protection agreement. This does not alter our adherence to PLOS ONE policies on sharing data and materials.

## 1. Introduction

Albumin is a non-glycosylated plasma protein widely distributed in mammals and synthetized exclusively in the liver been responsible for transporting metabolites, nutrients, hormones and drugs. Furthermore, it is also the main protein responsible for maintaining oncotic pressure within blood vessels [1]. Albumin is a 67 kDa multi domain globular protein stabilized by the union through 17 disulfide bridges and a helical shape, negatively charged facilitating its function to capture ligands and distributed in the blood plasma and maintaining oncotic pressure [2]. Equine albumin shows a 76.1% similarity with human albumin [3], specifically, equine albumin is composed of 583 amino acids, 2 amino acids less than human albumin [4].

In horses, the hypoalbuminemia is characterized by the decreased quantification of serum albumin concentrations below 2 g/dL [5]. When albumin is around or below this value, the oncotic pressure within the blood vessels is altered and some clinical changes such as muscle weakness, weight loss and generalized edema may occur [5]. In addition, the decrease in plasma albumin concentration below the range can generate other clinical damage due to an alteration in oncotic pressure, which can lead to ligands not being transported adequately, which can also limit the ability to purify compounds harmful to the body [6].

In Colombia there is not any scientific or clinical report available describing the use of equine albumin in horses. In clinical cases where albumin is required, like hypoalbuminaemia, anterior enteritis or diarrhea, veterinarians drawn upon the use of complete plasma or the use of human albumin, with the contamination risks and transference of red blood cells. The only available reports in our region comes from Brazil and Costa Rica, where researchers evaluated the administration of albumin to recover dehydrated horses and hyperimmune plasma producer horses respectively. In Brazil, Belli *et al*. analyzed the effect of administrating 8 mL/kg of a 5% solution of albumin in dehydrated horses. Their results showed the oncotic action of the exogenous albumin, increasing the fluid shift to the intravascular space [1]. Additionally, their results suggested that the use of albumin solution was preliminary, given that no side effects were observed. Despite the amount of albumin administrated, they obtained a large increase in serum albumin concentration due to the greater intravascular fluid retention and the albumin solution concentration [1]. Huertas *et al*. in Costa Rica, reported the adverse reactions after administrating 2 g/kg of an albumin solution into 6 horses destinated to produce hyperimmune plasma. The adverse reaction documented include generalized rash, deep and labored breathing, and gastrointestinal disturbances [7].

Following the International Conference on Harmonisation of Technical Requirements for Registration of Pharmaceuticals for Human Use (ICH) and the World Health Organization Standards to Manufacture Medicaments (WHO/MM), identifying patient safety as the main criteria, this study is aimed to analyze the clinical response in a reduced group of horses before using larger individuals [8–11]. It was very important to consider that there is not any previous experience in Colombia assessing the use of equine albumin in horses, also that the reported doses by Belli *et al*

and Huertas *et al* shows a wide range of albumin concentration administrated, and finally there is not any veterinary clinical guide indicating a safe concentration to be administrated. In order to design a high quality veterinary pharmaceutical formulation, this research was focused on the equine albumin pharmaceutical obtention/formulation for veterinary use and the further preliminary clinical response analysis, in adult horses, evaluating the main clinical and paraclinical parameters.

## 2. Materials and methods

### 2.1. Animals and experimental design

A total of 6 adult healthy Colombian Creole horses were used, 4 females and 2 males, 3 for blood obtention (named A♂, B♀ and C♀) and 3 for albumin administration (named 1♂, 2♀ and 3♀), that lived under the same sanitary conditions, grazing, feeding and coexistence in a herd. During the administration of albumin, an attempt was made to maintain the usual conditions of the animals, to do so, some samples collect and medical follow-up were carried out in the pasture. The animals used in the study were examined before, during and after the administration of the albumin formulation. Routine medical check-ups were performed to determine the values basic physiological characteristics and their variables in the period prior to treatment. All data were recorded in the individual medical record of each horse.

### 2.2. Ethics statement

All animals used in this study were maintained and treated under strict ethical conditions following the WHO for the production, control and regulation of snake antivenom immunoglobulins [12,13]. In addition, our protocols were approved by the Ethics Committee for the Use of Animals in Research CEEA of the University of Antioquia by protocol number N°155 of 03 October 2023.

### 2.3. Obtaining equine plasma

The plasma was obtained by extracting whole blood from horses healthy (horses A♂, B♀ and C♀). A volume of blood was extracted according to the weight of the donor animal and the status of different blood variables, according with the description WHO guidelines [12,13]. The blood was obtained by venipuncture of the external jugular vein through a cannula which was incorporated into a closed sterile bag system with anticoagulant. After obtaining the blood, the bag was remained at rest for 24 hours at 2–8°C allowing the decantation of red blood cells and the further separation of plasma. Once separated, and through pressure, plasma extraction is carried out into an empty sterile bag which is connected to the system.

### 2.4. Obtaining equine albumin

Obtaining, separating and purifying albumin involved a "Salting out" method with a series of successive filtrations using nominal filters and an ultrafiltration process, following the previous method described by Estrada *et al*, 2022 [14]. Once the final product was obtained, under a sterile environment, the product was filtered using a 0.22 μm filter under a sterile atmosphere for subsequent packaging and sealing.

### 2.5. Standard Quality Analysis of the albumin formulation

Based on the quality by design (QbD) principles described by the ICH Q8 R2 guidelines, sterility and hemotropics were classified as critical quality attributes (CQA) of our drug product (DP) since those attributes can directly affect patients safety [7].

#### 2.5.1. Albumin appearance and pH.
The appearance was determined through visual inspection using at least 10 vials. Black and white surfaces were employed to contrast the product after final sterile package, and the presence of any external particle in the liquid solution was carefully analyzed and recorded before the product was released for further

administration into the horses [15]. Finally, the albumin pH was determined using a calibrated pH meter (Orion model Star A221, Thermo Scientific. Waltham, MA, USA) at room temperature (26 ∘C). This parameter was assessed based on the 10 used vials of the product.

**2.5.2. Total protein concentration.** The total protein concentration of the albumin formulation was determined by the Biuret method, using a specific kit for quantification of total proteins (Reference 11500, Lot 41203, Biosystems, Barcelona, Catalonia, Spain) [12,14,8]. Briefly, 750 μL of Biuret reagent was added to 250 μL of an albumin sample. The mixtures were gently mixed and incubated for 30 min at 37 °C. After incubation, the absorbance was measured at 540 nm. Additionally, a calibration curve was carried out using known concentrations of Bovine Serum Albumin certified standard (BSA, Reference 11698, Lot 072XA, Biosystems, Barcelona, Catalonia, Spain) from 0.1 mg/mL to 50 mg/mL under the conditions described. The equation of the line was calculated for the concentration/absorbance relationship, and with this, we obtained concentration by interpolation. For this, the Prisma v 6.0 program was used.

**2.5.3. Total albumin concentration.** To determine the concentration of albumin in the formulation a specific kit for the detection of albumin based on a reaction with green bromocresol (Albumin Reagent, Reference 11547, Lot 46890, Biosystems, Barcelona, Catalonia, Spain) was used [14,8]. Briefly, 10 μL of each albumin (after reconstitution using 10 mL of WFI) were mixed with 1 mL of albumin reagent. The mixtures were gently shaken, and the absorbance was measured at 630 nm. Furthermore, it performed a calibration curve using the BSA standard certificate (BSA, Reference 11698, Lot 072XA, Biosystems, Barcelona, Catalonia, Spain) in concentrations from 1 mg/mL to 50 mg/mL working under the conditions previously described. The equation of the line was calculated for the concentration/absorbance relationship, and with this, the concentration was obtained by interpolation. For this, the Prisma v6.0 program is used.

**2.5.4. Purity and integrity.** To determine the purity and integrity of the albumin formulation, an electrophoresis using 12% polyacrylamide gels (SDS-PAGE) was performed under non-reducing conditions following methods for hemoderivates based on Laemmli methodology stained with the gel with FastGene Q-Stain (Cat FG-QS1) [12,14,8]. Albumin was loaded at a concentration of 1 μg/μL and a final volume of 20 μL, and a Precision Plus Kaleidoscope (Bio-Rad) was used as a standard for estimating MWs with markers covering the mass range from 250 kDa down to 10 kDa (Precision Plus Protein Kaleidoscope, Reference 1610375, Lot L001649B, Bio-Rad Laboratories, Hercules, California, United States).

**2.5.5. Molecular size distribution.** The molecular size distribution and purity of all albumin formulation batches was determined using an Agilent (Santa Clara, CA, United States) chromatograph with an internal standard of Human Albumin and a Shimadzu LC-40 chromatograph (Kyoto, Japan) with an external standard of Bovine Serum Albumin. In both cases we used a size exclusion column (SEC 3000, Phemomenex, Aschaffernburg, Bavaria, Germany) with an isocratic mobile phase in a $Na_2HPO_4$ buffer 15 mM, 30 Mm $NaH_2PO_4$ and 200 mM NaCl, with a flow rate of 0.5 mL/min, and a detection at 215 nm and 280 nm. Albumin and external standard samples were loaded using a concentration of 2 mg/mL in a 200 μL loop [12,8].

**2.5.6. Sterility and stability tests.** The sterility test was performed following the 71st Chapter of the United States Pharmacopeia through a third party laboratory certificated in Good Laboratory Practices (GLP). Shortly, albumin samples were added to the culture medium and was maintained for 14 days in incubation. Two types of growth media were used, i.e., fluid thioglycolate medium (FTM) and soybean casein digestion medium (SCDM). Associated with this same methodology and evaluating only the sterility parameter over time, the formulation stability was evaluated for 6 months after stored under refrigeration at 4° C [12].

**2.5.7. Hemotropics detection.** Following the ICH and WHO/MM principles, hemotropics analyses were performed through a third party laboratory certificated in Good Laboratory Practices (GLP). Shortly, samples from albumin batch AL-15052023 were analyzed and subjected to a polymerase chain reaction in real time (PCRrt). The PCRrt assay for the heamoplasmas detection was performed using an internal validated protocol with a set of universal primers for hemotropic *Anaplasma sp*, *Babesia sp*, *Mycoplasma sp*, *Trypanosoma sp* and *Theileria sp*.

## 2.6. Albumin administration

Two 14G conventional venous catheters were installed each animal (horses 1♂, 2♀ and 3♀), one on the right jugular for the administration of albumin, and another on the left jugular for taking samples and administration of emergency medications if necessary. A total of 525 mL of each albumin solution was administered adjusting and ensuring a total of 5334 mg of albumin (10.16 mg/mL). After the administration, the animal was closely monitored for 2 hours to obtain blood samples and evaluate the clinical parameters (see below). In order to simulate the usual environmental and behavioral conditions for the animal, all three individuals were taken back with the herd with food and water *ad libitum* where more samples and clinical data were obtained (see below). After the 24 hours, the individuals were constantly evaluated by our veterinary team for a maximum of 72 hours.

## 2.7. Clinical examination

The animal was evaluated with the support of Equine practitioners, before during and after treatment, the following variables were evaluated in the clinical examinations: Mental status, mucous membrane status, capillary refill time, return time of skin fold, jugular filling time, blood pressure, heart rate, frequency respiratory rate, hydration status, presence or absence of digital pulses, body temperature, abdominal auscultation. All data results were recorded in a table and the respective medical record.

## 2.8. Paraclinical analysis

Venous blood samples were taken just before the administration of albumin (t=0) and during the following postadministration times: 2, 4, 8, 16, 32 minutes, and 1, 2, 4, 8, 12, 18 and 24 hours. To do so, we used 3.5 ml tubes, without anticoagulant or with EDTA. Samples were sent to an external clinical veterinary laboratory (VITALAB®), where they were processed to quantify: hemoleukogram, albumin concentration and differentiated plasma proteins (aspartate aminotransferase) AST, (Blood urea nitrogen) BUN, urea concentration, creatinine concentration, (Gamma-glutamyl transferase) GGT, sample processing was performed using a Mindray BA88a device.

## 2.9. Measurement of blood gases, electrolytes, lactate, hematocrit and acid-base status

Using the portable blood analyzer Epoc Blood Analysis® system, the gases, ions and metabolites, were measured by using a drop of venous blood obtained in a syringe with lithium heparin furtherly added in the equipment cartridge. The variables measured were: pH, partial pressure of $CO_2$ ($pCO_2$), partial pressure of $O_2$ ($pO_2$), sodium ($Na^+$), potassium ($K^+$), calcium ionized ($iCa^{++}$), glucose (Glu), lactate (Lac), hematocrit (Hct) and temperature (T°). The equipment also determined: hemoglobin (cHgb), bicarbonate ($cHCO_3$-), carbon dioxide total (cTCO), blood base excess (BE), extracellular base excess (BE) and oxygen saturation (cSO). They were measured in three moments: before, during and at the end of the treatment in venous blood.

## 2.10. Measurement of blood pressure and heart rate variability analysis

Non-invasive blood pressure were measured using the veterinary blood pressure monitor SunTech Vet25E, installing the cuff at the base of the horse tail. The measurements were obtained before, during and after albumin administration. To measure heart rate, a Polar H10 sensor was used, the results were measured using the Polar equine application App and interpreted using Kubios HRV lite software.

## 2.11. Statistical analysis

Epoc Blood Analysis® results and laboratory results were recorded on a spreadsheet calculation, an ANOVA test was applied followed by a Tukey post-test ($p < 0.05$). The analysis Statistics were carried out with the help of Prisma v6.0 software.

## 3. Results

### 3.1. Albumin production

A total of 11.03 liters of blood was obtained from all 3 horses without any effect over the donor horses. After the 24 hours of cold storage of the blood in the sterile bags with anticoagulant a total of 7.8 liters of plasma was obtained with a pH average of $6,66 \pm 0,18$ and a conductivity average of $11,66 \pm 1,05$ ms/cm (Table 1).

A total of 3 different batches of albumin was manufactured from the plasma obtained from each horse. From all three batches and after processing the albumin, a total of 2800 mL of bulk product was obtained. At the end, afterwards filtration through a 0.22 µm filter and the subsequently filling using sterile 10 mL and 50 mL vials, a total of 169 units were obtained (Table 2). After each albumin batch was sterile filled, the albumin was identified with a batch number consisting of the initial letter of the product followed preparation date (AL-DDMMYY). Each vial was subsequently labeled for further quality analysis (Fig 1).

At the end of each batch, a yellow product was obtained, with no suspended visible particles or turbidity or precipitated (Fig 1 A and B). The pH of each batch was 6,84, 6,86, and 6,78.

### 3.2. Standard quality characterization of the albumin formulation

When quantifying the concentration of total protein using the Biuret method, each albumin batch revealed different protein concentration according to the concentration method during the tangential filtration. The first batch showed the lower concentration while the last two showed the higher (and similar) concentrations (Table 3). In all cases, the calibration curves used in the linear extrapolation model presented an $r^2$ above 0,99. When quantifying the concentration of total albumin using the specific green bromocresol method, the results showed that we were able to obtain an albumin concentration ranging from 61% to 78% (Table 3). In all cases, the calibration curves used in the linear extrapolation model presented an $r^2$ higher than 0,99.

### 3.3. Electrophoresis

The electrophoresis performed to each final batch seeking to determine the integrity and the purity of the albumin showed an important and predominant band with a molecular weight above 50 kDa that may correspond to the separated and purified albumin (Fig 2). Below the 50 kDa no important bands were detected, indicating the integrity of the protein and the

**Table 1. Amount and percentage of plasma obtained from each healthy donor horse following the World Health Organization methodology [8].**

| Horse/ sex | Collected blood (mL) | Obteined plasma (mL) | % Yielded | Conduct (ms/cm) | pH |
|---|---|---|---|---|---|
| A ♂ | 3310 | 2260 | 68,27% | 11,36 | 6,85 |
| B ♀ | 5020 | 3760 | 74,90% | 12,84 | 6,67 |
| C ♀ | 2700 | 1832 | 67,85% | 10,8 | 6,48 |

**Table 2. Batches of albumin manufactured including the number of final vials obtained after the sterile filling of each vial.**

| Batch number | Bulk product volume (mL) | Units of vials |
|---|---|---|
| AL-15052023 | 1400 | 12 vials 50 ml<br>40 vials 10 ml |
| AL-02042024 | 600 | 58 vials 10 ml |
| AL-10042024 | 800 | 71 vials 10 ml |

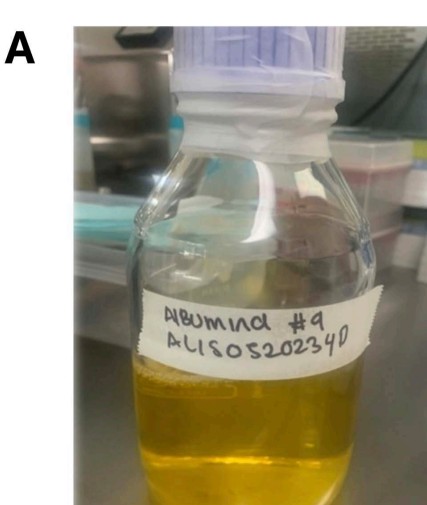

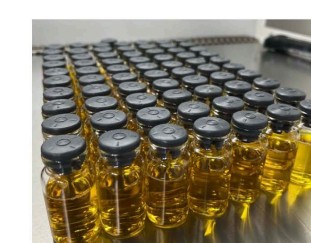

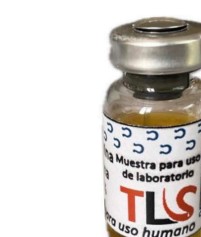

**Fig 1. Albumin batches obtained by the salting out method followed by a nominal and tangential filtration after filling and labeling.** A: Bulk product. B: Final product after sterile filling. C: Final product after labeling.

**Table 3. Protein and albumin concentration across each final product batch obtained by the salting out method followed by a nominal and tangential filtration and after sterile filling.**

| Batch | Protein concentration (mg/mL) | Albumin concentration (mg/mL) | % of albumin yielded |
|---|---|---|---|
| AL-15052023 | 12,91 | 10,16 | 78,69% |
| AL-02042024 | 56,21 | 34,73 | 61,34% |
| AL-10042024 | 40,57 | 26,19 | 64,55% |

method suitability to obtain the complete protein. Additionally, other bands were detected which may correspond to different plasma proteins with molecular weights similar to those like ceruloplasmin (132 kDa) and other plasma content.

### 3.4. Chromatography

The size exclusion chromatography performed to each one of the final product batches showed a majority and predominant fraction corresponding to albumin eluting at 19,51 minutes along with other minor components that elute at different times. Fig 3 shows batch AL-15052023 run compared against an internal standard of human albumin (HA). The major fraction showed a retention time similar to the HA monomer internal standard with a relative abundance above 89% (Fig 3 A and Supplementary table S1 and table S2 in S1 File). Further SEC-HPLC runs, under the same conditions as previously described, but using Bovine Serum Albumin (BSA) as an external standard showed a similar result, where a predominant fraction was detected with a similar retention time around 13,2 minutes. It is notable that the standard showed an additional fraction eluting at 22,1 minutes (Fig 3B). Fig 3 C to E shows a consistency un the retention time of the major fraction directly related with the albumin presence as observed in the electrophoretic profile.

### 3.5. Sterility and stability tests

Sterility tests, performed in a GLP certified laboratory, showed that all the final products, from all the three batches of the equine albumin did not present any type of contamination for the microorganisms evaluated (gram negative bacteria, gram positive or fungi). Original laboratory reports are available in the supplementary material (Figure S1 in S1 File). In

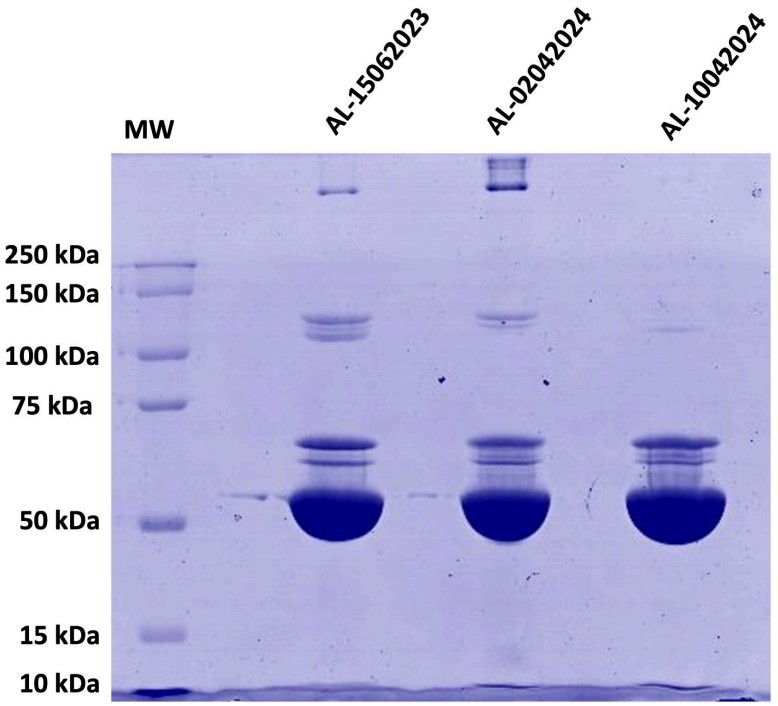

**Fig 2. 12% SDS-PAGE electrophoresis of each final albumin batch a concentration of 1µg/µL and a final volume of 20 µL of albumin and sample buffer under non-reducing conditions.** Precision Plus Kaleidoscope (Bio-Rad) was used as a standard for estimating MWs with markers covering the mass range from 250 kDa down to 10 kDa.

total, 5 samples from batch AL-15052023 were sent from in consecutive months after manufacturing. In all samples the result was: "absence of growth". Therefore, the sterility of the final product is guaranteed. Batches AL-02042024 and AL-10042024 were analyzed immediately after batch sterile filling and labeling and just before administration (30 days after manufacturing). In all cases the result was: "absence of growth".

### 3.6. Hemotropic tests

Sample from batch AL-15052023 showed absence of *Anaplasma sp*, *Babesia sp*, *Mycoplasma sp*, *Trypanosoma sp* and *Theileria sp*. Original laboratory reports are available in the supplementary material (Figure S2 in S1 File).

### 3.7. Albumin administration

After all the Standard Quality Characterization to make sure that the albumin formulation was suitability to be administrated into the three horses, a final amount of 5334 mg of albumin, from all 3 batches, were administrated in a final volume of 525 mL in a concentration of 10.16 mg/mL. The time of administration in all cases took around 22 minutes.

 **3.7.1. Clinical variables.** No significant adverse clinical changes were observed after the IV administration of the albumin formulation. Supplementary table S3 in S1 File shows all the measured variables with no clinical significance deviation. In all 3 cases the animals remained active, alert, responding to external stimulus, with no changes in their mental status or mucosa's. The capillary refill time, heart and respiratory rates stayed in the range (Supplementary table S3 in S1 File). The rectal temperature, systolic and diastolic pressure showed normal values. After digestive auscultation, horses remained normal gastrointestinal motility pattern and, in some cases, showed hypermotility, probably by the movement of fluids within the gastrointestinal tract in response to changes in osmotic pressure. Equine 1 showed

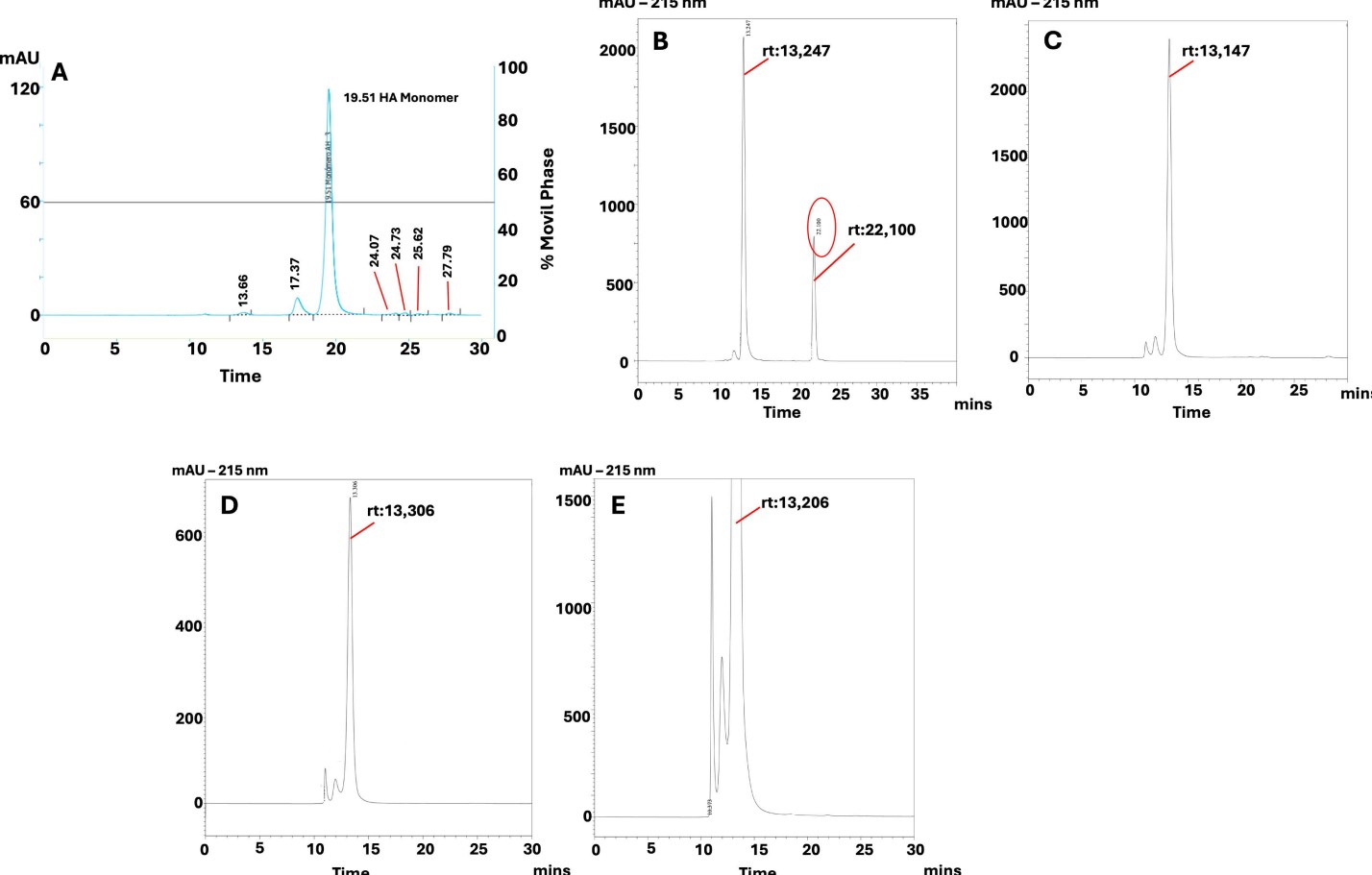

**Fig 3. Chromatographic profile of albumin. A:** SEC profile of albumin batch AL-15052023 using an internal Human Albumin validated standard in an Agilent chromatography equipment with a SEC 3000 Phenomenex column and a movil phase consisting in a Na$_2$HPO$_4$ 15mM, NaH$_2$PO$_4$ 30mM, NaCl 200 mM buffer, a 0.5 mL/min flux and a 215 nm. **B:** SEC profile of BSA external standard in a Shimadzu chromatography equipment with a SEC 3000 Phenomenex column and a movil phase consisting in a Na$_2$HPO$_4$ 15mM, NaH$_2$PO$_4$ 30mM, NaCl 200 mM buffer, a 0.5 mL/min flux and a 215 nm. **C to E:** SEC profile of albumin batches AL-15052023 **(C)**, AL-02042024 **(D)** and AL-10042024 **(E)** using an external BSA standard in a Shimadzu chromatography equipment with a SEC 3000 Phenomenex column and a movil phase consisting in a Na$_2$HPO$_4$ 15mM, NaH$_2$PO$_4$ 30mM, NaCl 200 mM buffer, a 0.5 mL/min flux and a 215 nm. rt: retention time. Albumin and external standard samples were loaded using a concentration of 2 mg/mL in a 200 μL loop.

positive pulses in the left posterior extremity, while in equine 3 icteric sclera was detected. Fig 4 and Table 4 shows heart rate fluctuation graphical representation from each equine and the heart rate variability time domain analysis including parasympathetic index.

**3.7.2. Physiological analysis.** Physiological changes were observed in response to the administration of albumin, which are related to its function as a plasma expander. When calculating the change in plasma volume after albumin administration, it is estimated that each animal received approximately six liters of fluid in the intravascular space [9]. This fluid redistribution, driven by the colloid-osmotic properties of albumin, may have caused a transient hemodilution, reflected in the initial increase in hematocrit, which subsequently stabilizes at basal levels due to the horse's own redistribution mechanisms.

**3.7.3. Paraclinical analysis.** Paraclinical results were divided in four different lines to facilitate the analysis: gases and electrolytes, red blood component, white blood component, general blood proteins and other metabolites. Samples were

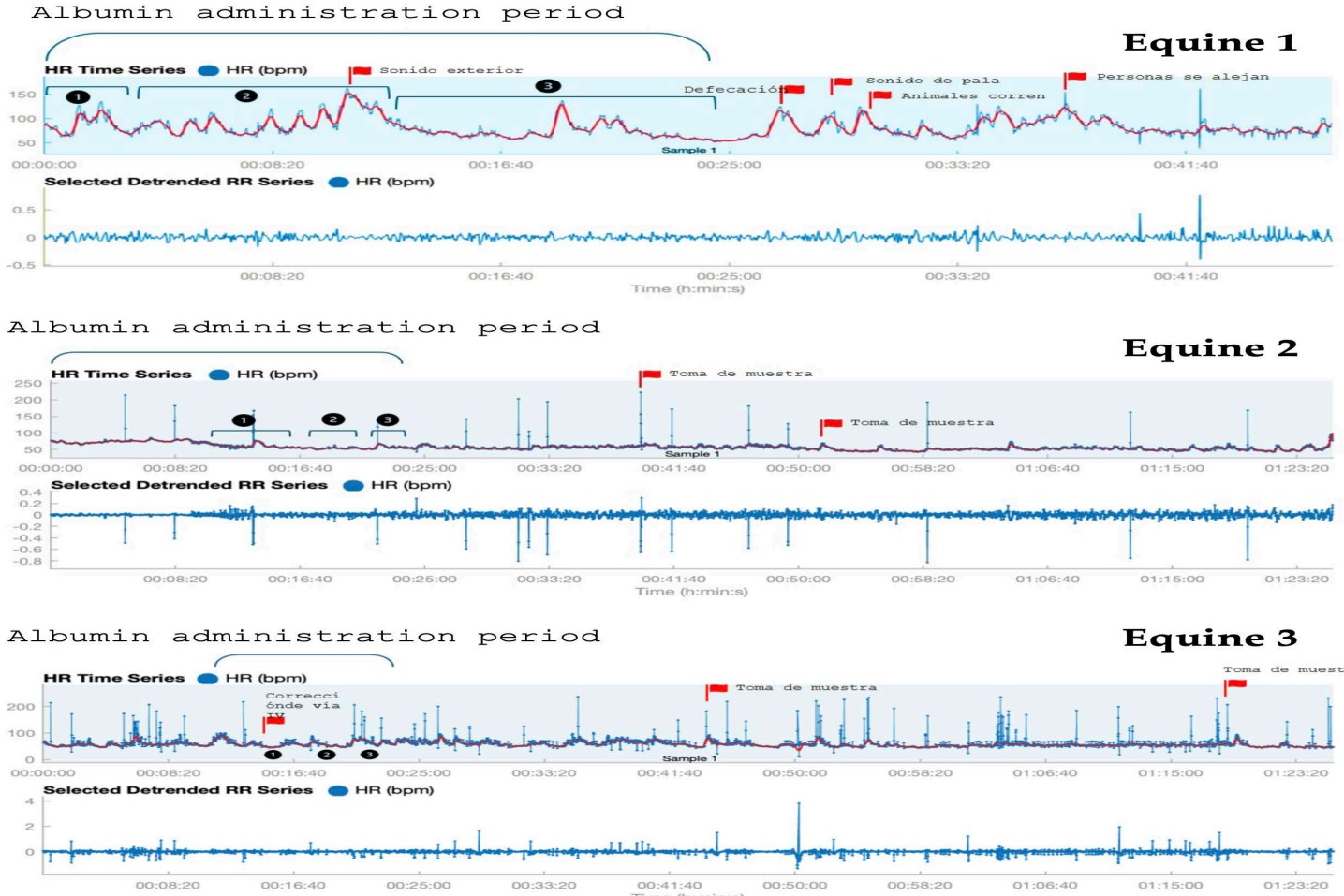

**Fig 4. Heart rate fluctuation graphical representation from each equine, registered before, during and after the albumin administration.** Red flags indicates any significant change.

**Table 4. Heart rate variability time domain analysis including parasympathetic index. HR min: minimal cardiac frequency. HR max: maximal cardiac frequency. HR prom: average cardiac frequency. RRmean: average duration in milliseconds (ms) between cardiac beats (R wave interval). RMSSD: parasympathetic tone indicator. PNSindex: parasympathetic index. SNSindex: sympathetic index.**

| Animal | HR min (bpm) | HR max (bpm) | HR prom (bpm) | RRmean (ms) | RMSSD (ms) | PNSindex | SNSindex | Duration (min) |
|---|---|---|---|---|---|---|---|---|
| Equine 1 | 52 | 162 | 87 | 731 | 35 | −1,11 | 0,45 | 46 |
| Equine 2 | 43 | 104 | 59 | 1049 | 56 | 1 | −1,26 | 1h20 |
| Equine 3 | 31 | 141 | 60 | 1025 | 196 | 4,75 | −1,8 | 1h25 |

taken before the albumin administration, during the administration and after the administration. Regarding the creatine kinase (CK), gases and electrolytes, we were not able to record the data from equine 3 during the albumin administration since we had an issue with the Epoc Blood Analysis® cassette and the CK sample in the analysis laboratory.

Blood gases, electrolytes, lactate, hematocrit and acid-base did not showed any significant changes (Supplementary table S4 in S1 File). Red blood line showed a slight diminish in the values during the 10 minutes post-administration,

which where observer to be normalized after the minute 20 until minute 100 where another alteration was observed. In all cases the values remained on range (Fig 5).

White blood cells line count showed specific changes related with the redistribution of platelets in the vascular bed after the albumin administration. Leucocytes, neutrophils and eosinophils a similar pattern was observed with some specific changes during the first 20 minutes post-administration. An increase in the concentration of neutrophiles, lymphocytes and leucocytes after 8 hours post-administration. As seen in the red blood line count after minute 100 where another alteration was observed. In all cases the values remained in the rank (Fig 6).

Similar to the red blood line count, proteins showed a slight diminish in the values during the 10 minutes post-administration, which where observer to be normalized after the minute 20 until minute with a further alteration after minute 100. Fibrinogen did not present any significant change after the albumin administration. In all cases the values stayed in the rank (Fig 7).

Aspartate aminotransferase (AST) and creatinine kinase (CK) did not presented any significant change after the albumin administration (Fig 8).

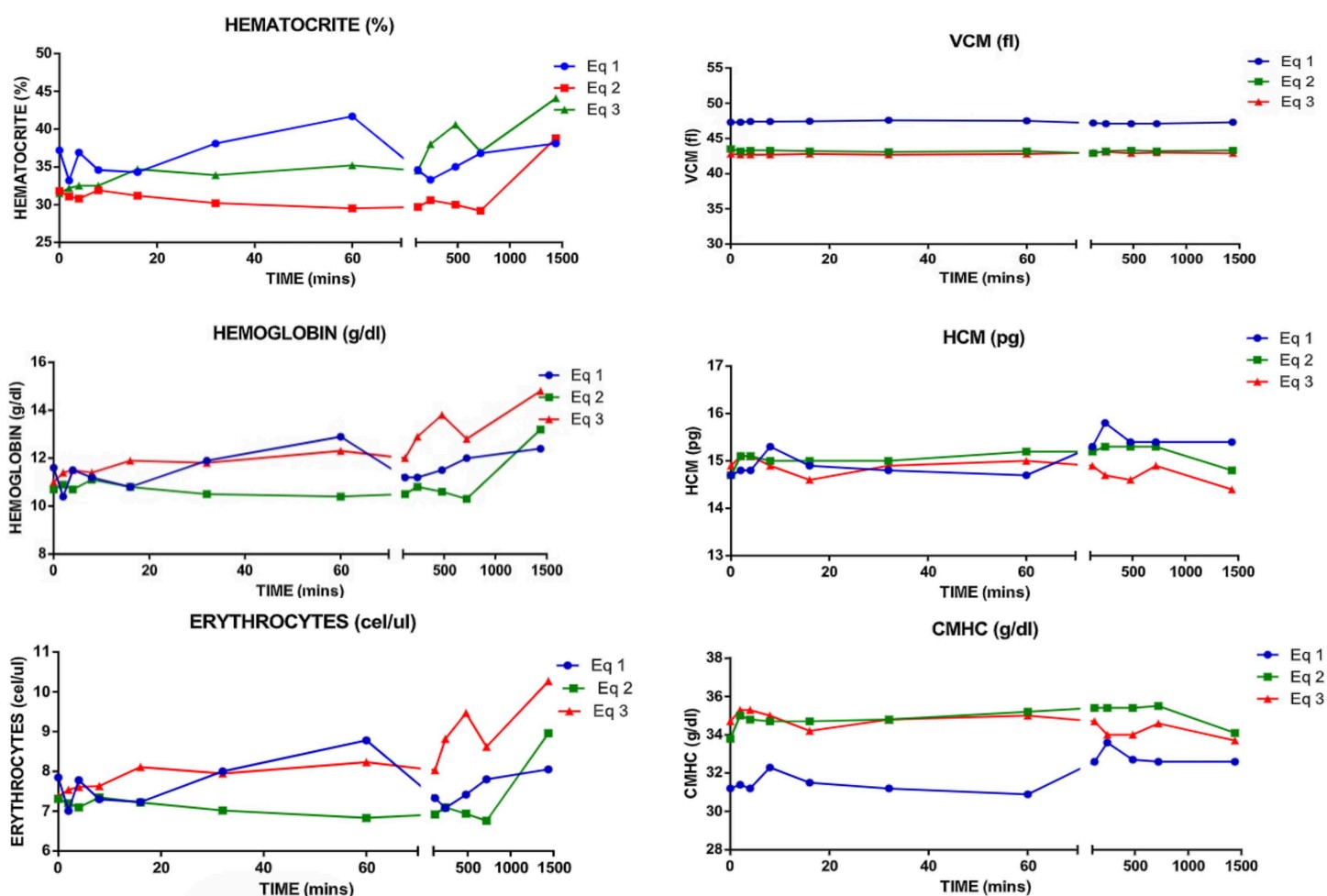

**Fig 5. Red blood cell line count post-administration of albumin.** VCM: mean corpuscular volume. HCM: hemoglobin corpuscular volume. CMHC: average corpuscular hemoglobin concentration.

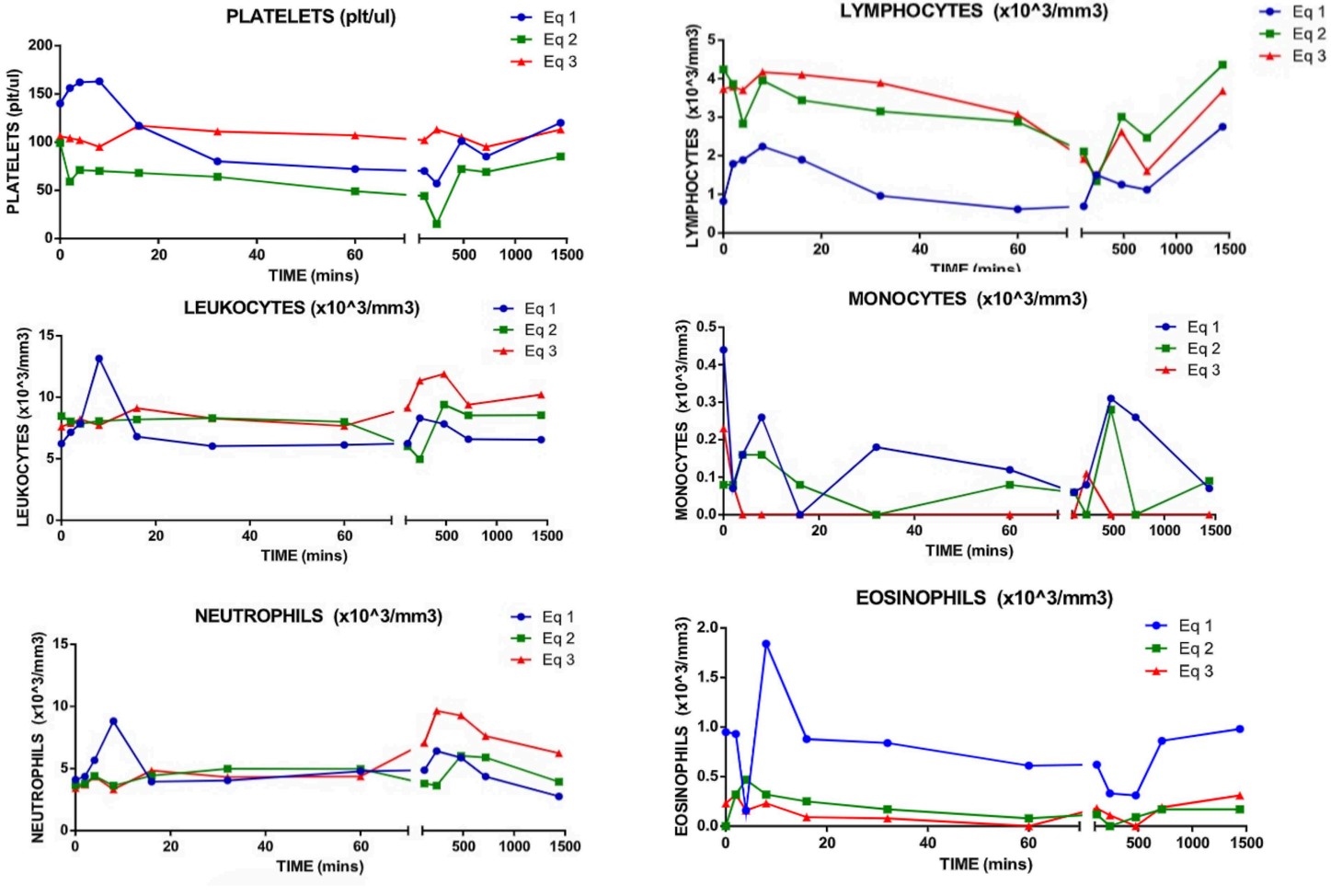

**Fig 6. White blood cell line count post-administration of albumin.**

## 4. Discussion

Currently, in Colombia there is not available a specific albumin equine pharmaceutical preparation formulated to treat diseases associated with acute abdomen syndrome that can lead to dehydration, hypovolemia, hypoalbuminemia and hypoproteinemia.

The proposed bleeding method with the further system to obtain plasma, allowed the obtention of the necessary plasma to be processed and obtain the albumin as previously described by other authors to obtain hyperimmune plasma [10]. It is important to mention that the horses did not showed any local or systemic physical alteration after the blood donation. The albumin obtention method allowed the separation and purification of sterile albumin, with the required quality attributes to be administrated intravenously fulfilling the assayed standard quality characteristics described by the WHO guidelines as previously described by Estrada *et al*. and Tabarez *et al* [14,8]. At the end of the process, an sterile formulation of equine albumin was obtained with a high purity level above the 60% (and up to 78%) and a total protein concentration up to 36.56 mg/mL. The purity and concentration variability observed in the batch number AL-15052023 regarding the further two batches corresponds to the standardization process where a higher concentration during the tangential filtration were performed to improve the final concentration of the albumin formulation. Based on the QbD principles, CQA

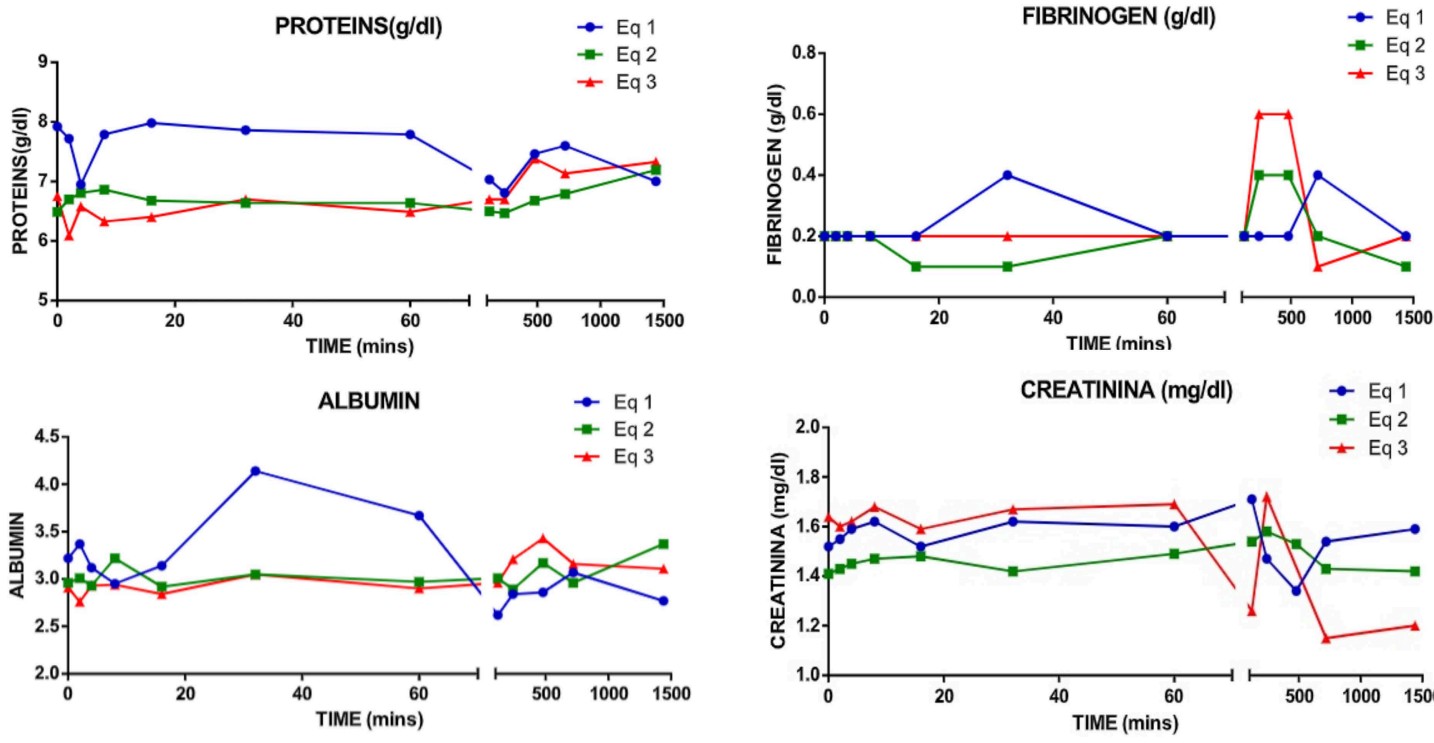

**Fig 7. General protein line post-administration of albumin.**

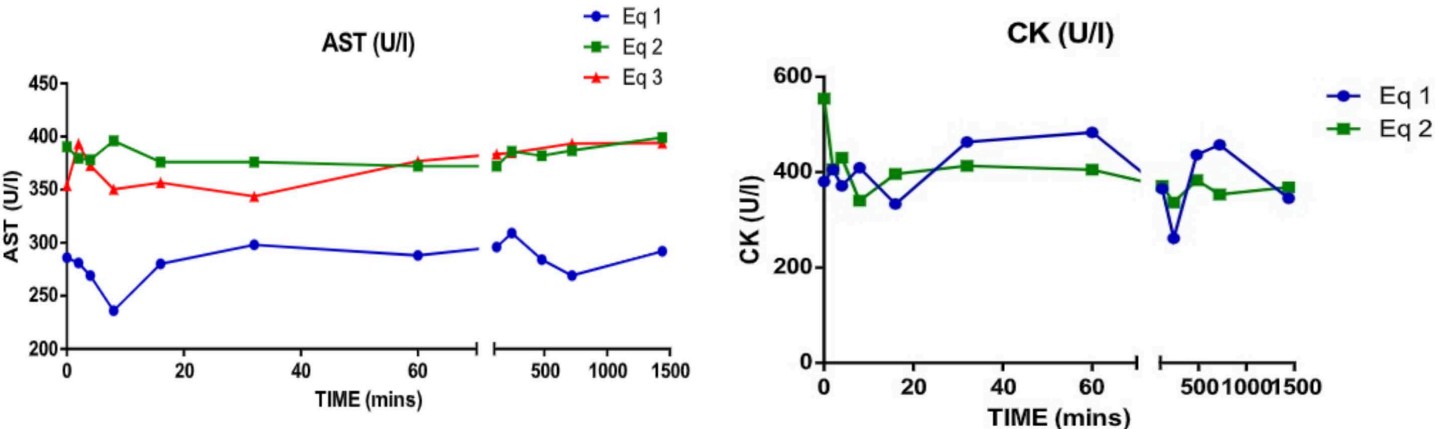

**Fig 8. Other metabolites line quantification post-administration of albumin.** AST: aspartate aminotransferase. CK: creatinine kinase.

correspond to DP characteristics directly identified and based on the severity of harm to a patient based on safety and efficacy [11]. In our process, tangential and sterile filtration were identified as critical process parameters (CPP), and the respective attribute as a CQA (sterility and hemotropics). These CQA were analyzed by certified laboratories with validated procedures (see below) since these attributes were essential to ensure patients safety and product high quality. Sterility CQA parameter on the albumin formulation showed absence of any bacterial, fungi or hemotropic parasites. This

is absolutely important to guarantee a basic safe product to be administrated specially with hemotropics since they can lead to infections involving anemia and mortality in mammals [11]. The analyses by external certified laboratories, with validated methods, were essential to ensure a high quiality albumin formulation.

After the albumin administration slight alterations, without compromising the healthy conditions in all 3 horses (equines 1, 2 and 3), were detected in the paraclinical analysis. When evaluating blood pH, no alterations were found, which was expected given that these were healthy animals; however, studies in humans have indicated alkalization of pH in patients with metabolic acidosis [1]. Regarding electrolytes, only a decrease in potassium was observed in horse 1, horse 2 and horse 3, without this being reflected in cardiological alterations. Initial blood lactate in horse 2 was elevated, which could be attributed to the horse's physical activity in the paddock, since before the study, it was running, activating the anaerobic metabolic pathway. However, over time, lactate values decreased, while in the other individuals they did not go out of range. The missing data results during the blood gas analysis from horse number 3 corresponds to difficulties in processing the sample during the albumin administration. To describe the changes in serum albumin, it is important to understand that, in the used healthy animals horses, the administration of albumin can generate protein catabolism, this is why with exogenous albumin and due to its functions to expand plasma volume, serum albumin concentrations decrease in the first 20 minutes after administration and after this time the levels increase until the first hour after administration when these values are restored as previously described by Belli *et al*. [1]. Additionally, as previously described by Huertas et al. we did not expect a rise in the albumin concentration in the early-stage post-administration due to the greater intravascular fluid retention and the albumin solution concentration [16]. The red blood cell line analysis (hematocrit, hemoglobin and erythrocyte count) showed a slight decrease in the first ten minutes after administration, possibly due to a transient hemodilution caused by the plasma volume expansion properties of albumin [9,15]. However, from 20 minutes after administration, the levels began to increase, possibly in response to the increase in oncotic pressure induced by albumin, subsequently stabilizing at levels similar to the initial ones. No significant changes were observed in mean corpuscular volume, mean corpuscular hemoglobin and mean corpuscular hemoglobin concentration. Regarding platelet count, changes related to platelet redistribution were recorded after albumin administration. The most notable changes were observed in horse 1, which had an initial count of 140 platelets/μL, which decreased to 72 platelets/μL during the first hour after administration. When comparing the three horses, a similar trend was observed: after two hours of administration, platelet counts decreased and stabilized until eight hours. It is believed that this decrease is achieved by the binding of platelets to exogenous albumin.

When reviewing the use and availability of albumin in different species, multiple studies in canines are observed, where better results in terms of safety and efficiency are reported in species-specific human albumin versus heterologous albumins. Trow *et al* showed, in an extensive review, how albumin was used in dogs. A typically 25% albumin solution is commonly dissolved in 10% solution with saline (0.9% NaCl) and administered through a transfusion filter [17]. The most frequent indication corresponds to septic peritonitis, neoplasia, lymphoma, severe trauma, hepatic disease, hypoalbuminemia, pancreatitis, among others. The Median dosage of human albumin administered was 1.4 g/kg [15,17]. Despite the difference in albumin concentration used in our study (0,014 g/kg), our main focus was to determine, in a preliminary dose, any adverse effect after administration.

When analyzing leukocyte, neutrophil and eosinophil counts, similar patterns were observed in the three equines, with changes attributable to handling stress in the first 20 minutes after albumin administration. However, no significant alterations were recorded in monocytes, whose values remained within the normal range. Lymphocytes decreased until two hours after administration, after which they began to gradually recover. Studies by Casulleras *et al* and Alcaraz-Quiles *et al*. attribute the immunomodulatory function to human albumin, where they describe that albumin is internalized within the cytoplasm of leukocytes and interferes with the signaling of Toll-like receptors [18,19]. The increases in neutrophils, lymphocytes and leukocyte counts 8 hours after albumin administration may be associated with the fact that exogenous oxidized forms of albumin can promote a proinflammatory cytokine storm; therefore, it is likely that contact of the albumin

solution with ambient air during its preparation produced this effect [18]. However, the supposedly inflammatory effect was not sustained during the observation period.

When evaluating the total protein concentration, a decrease was observed in the first 8 minutes after administration, possibly as a result of a transient hemodilution. However, after 16 minutes, the protein concentration increased and remained stable without relevant changes. No significant alterations were recorded in the levels of fibrinogen and globulins throughout the study. CK results, samples from horse 3 were not processed, so the analysis is limited to horses 1 and 2. In these, AST and CK levels did not show significant changes, while creatinine showed a variation from the first hour after albumin administration, indicating adequate renal function. This phenomenon can be explained by the increase in glomerular filtration associated with temporary hemodilution, evidenced by the decrease in serum creatinine values, which produced thirst in the animals, so when left in the paddock they immediately looked for water. In human studies, it has been shown that patients with kidney failure present a decrease in creatinine levels after albumin administration [1,20].

HRV analysis showed that parasympathetic tone was predominant, even external stimuli were detected, this confirm that albumin i.v. infusion did not promote sympathetic response (p.e. tachycardia).

Our study limits are focused on the number of individuals used to analyze the clinical effects of albumin administration. We initially used a reduce number of healthy individuals following the ethical committees' recommendations in our country, specially due to the lack of previous information in Colombian Creole Horses. This leaded us to describe this study as preliminar.

## 5. Conclusions

The methodology used allowed the manufacture of equine albumin fulfilling the Standard Quality Characteristics requirements for intravenous administration. Furthermore, its administration to healthy horses does not alter clinical or paraclinical parameters at the concentration used. This indicates the ability to obtain a product that can potentially be used as a pharmacological therapy in horses. These results can be a pilar for investigating and standardizing the use of specific equine albumin in sick horses.

It is important to continue with further studies including higher of individuals in order to establish safety and efficacy parameter, and protocols for the administration of albumin in its different therapeutic indications.

## Supporting information

**S1 File. Table S1.** Original result data tablesheet (in spanish) obtained from the chromatography software.; **Table S2.** Transcribed and translated data sheet results table obtained from the software (the original is available in the supplementary material). Highlighted data indicates the percentage of relative abundance of the fraction corresponding to human albumin (HA).; **Figures S1.** Original sterility assay result issued by the GLP certified laboratory (in spanish) indicating the sterility of A: Batch AL-15052023. B: Batch AL-02042024. C: Batch AL-10042024. **Figures S2:** Original hemotropic assay result issued by the GLP certified laboratory (in spanish) indicating the absence of *Anaplasma sp, Babesia sp, Mycoplasma sp, Trypanosoma sp and Theileria sp*. on batch AL-15052023. **Table S3.** Table results with the physiological constants including the arterial preasure measurment. Table abreviations: CRT: capillary refill time, SEC: seconds, CF: cardiac frecuency, BPM: beats per minute, RF: respiratory frecuence, BPM: breaths per minute, AAP: average arterial preasure, T: corporal temperature, M/S/P: moist, shiny pink, DP: dry and pink, PM: pale and moist, NA: not analyzed, NBS: normal bowel sounds, IBS: increase bowel sounds, LQ: left quadrant, URQ: upper right quadrant, ULQ: upper left quadrant, LHL: left hind limb. **Table S4.** Venous gases results in all three horses using the Epoc Blood Analysis® equipment. Ad: plasma administration. Horse three is missing the measure during plasma administration. Raw 12% SDS-PAGE electrophoresis results of albumin batch AL-15062023 at different concentrations.
(DOCX)

## Author contributions

**Conceptualization:** Sebastian Estrada-Gomez.

**Data curation:** Vanessa Cifuentes, Angélica Maria Zuluaga-Cabrera, Leidy Johana Vargas-Muñoz, Sebastian Estrada-Gomez.

**Formal analysis:** Vanessa Cifuentes, Angélica Maria Zuluaga-Cabrera, Leidy Johana Vargas-Muñoz, Sebastian Estrada-Gomez.

**Funding acquisition:** Sebastian Estrada-Gomez.

**Investigation:** Sebastian Estrada-Gomez.

**Methodology:** Vanessa Cifuentes, Angélica Maria Zuluaga-Cabrera, Sebastian Estrada-Gomez.

**Project administration:** Sebastian Estrada-Gomez.

**Writing – original draft:** Vanessa Cifuentes, Angélica Maria Zuluaga-Cabrera, Sebastian Estrada-Gomez.

**Writing – review & editing:** Vanessa Cifuentes, Leidy Johana Vargas-Muñoz, Sebastian Estrada-Gomez.

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
