## [Decision Letter · Decision Letter 0]

7 Oct 2025

Dear Dr.  Estrada-Gomez,

Thank you for submitting your manuscript to PLOS ONE. After careful consideration, we feel that it has merit but does not fully meet PLOS ONE’s publication criteria as it currently stands. Therefore, we invite you to submit a revised version of the manuscript that addresses the points raised during the review process.

Please submit your revised manuscript by Nov 21 2025 11:59PM.  If you will need more time than this to complete your revisions, please reply to this message or contact the journal office at plosone@plos.org . A rebuttal letter that responds to each point raised by the academic editor and reviewer(s). You should upload this letter as a separate file labeled 'Response to Reviewers'.A marked-up copy of your manuscript that highlights changes made to the original version. You should upload this as a separate file labeled 'Revised Manuscript with Track Changes'.An unmarked version of your revised paper without tracked changes. You should upload this as a separate file labeled 'Manuscript'.

We look forward to receiving your revised manuscript.

Kind regards,

Claudia Interlandi, Ph.D

Academic Editor

PLOS ONE

Journal Requirements:

4. Please note that funding information should not appear in any section or other areas of your manuscript. We will only publish funding information present in the Funding Statement section of the online submission form. Please remove any funding-related text from the manuscript.

Additional Editor Comments:

Reviewer 1

The study proposes the preparation and evaluation of intravenously administered albumin in horses.

Title: Does not reflect the study performed

The study objectives do not reflect the study performed.

The study focuses more on the production of albumin for commercial use. Its use in horses was tested on only three horses, which does not allow for any statistically significant conclusions.

The focus of the study, based on the title, should be on the preparation of this commercial albumin for horses. Administering it to three horses, performing hematological and clinical analyses, and suggesting the product's safety could be dangerous. Case reports can be reported, but they do not encourage its use.

The authors mention evaluating plasma volume expansion. How was this evaluated? I did not see any assessment of plasma volume expansion in the text that could be considered an appropriate methodology for this purpose.

Furthermore, the introduction is long and contains basic concepts. It needs to be more focused on horses. There is a lot of human data, and there are already studies on horses and albumin administration (lines 50-57; 44-46). Line 77 is missing the reference date. Lines 214-217 need to be written better. Line 247: non-invasive blood pressure.

If the animals' response to the administered product is to be evaluated, a phase 1 and phase 2 clinical trial would need to be conducted, following the steps and an adequate number of animals.

Therefore, the text needs to be adapted to the albumin production and administration reports, and then reevaluated.

reviewer 2

Dear authors, thank you for submitting a detailed report of a very well constructed and undertaken study.

The manuscript is overall well written with very few typological error and good grammar. However it is percievable that it is not written by an english native. i would suggest an english proof reading as some of the sentences have a litteral translation from the latine way of phrase construction and some of the word choicescould be improved. that would help with the overall comprehension of the manuscript for the reader.

Concerning the content of the manuscript, the M&M is well written and the methodology very thorough.

The results are well presented eventhough there are some discussion elements in this part that should be transfered to the dedicated section. With three horses only (which is a good start), it is difficult to interpret the overall blood parameters variations overtime. You have on one of the three horses blood parameters that seem to behave differently. I would be cautious about overinterpreting these results and a clear and detailled description would be more informative.

It should be stated that you did not document any clinical sign of immediate immune reaction to the albumin infusion. Didi you look for delayed reaction?

In the discussion, the limits of the study are not well described.

I would also encourage you to deepen the discussion using data published in other species like dogs.

Finally, I had a hard time understanding how you evaluated the plasma expansion following the infusion.

Apsect to consider that can interfere with the variation of the blood parameters are : did some of the horses got thirsty during the infudion, this is something we certainly see with the administration of hypertonic saline and if a horse decided to drink more than another, it would have an effect on the Ht and oncotic pressure / plasma volume. Consider also a potential splenic contraction that could explain a short lived increased hematocrit.

Concerning platelet counts and variations, did you use only the automat results or did you also check a blood smear?

Thanks again for the quality of the study, I'm looking forward reading your revisions

Reviewer's Responses to Questions

**Comments to the Author**

1. Is the manuscript technically sound, and do the data support the conclusions?

Reviewer #1: No

Reviewer #2: Yes

2. Has the statistical analysis been performed appropriately and rigorously?

Reviewer #1: I Don't Know

Reviewer #2: N/A

3. Have the authors made all data underlying the findings in their manuscript fully available?

Reviewer #1: Yes

Reviewer #2: Yes

4. Is the manuscript presented in an intelligible fashion and written in standard English?

Reviewer #1: Yes

Reviewer #2: No

Reviewer #1: The study proposes the preparation and evaluation of intravenously administered albumin in horses.

Title: Does not reflect the study performed

The study objectives do not reflect the study performed.

The study focuses more on the production of albumin for commercial use. Its use in horses was tested on only three horses, which does not allow for any statistically significant conclusions.

The focus of the study, based on the title, should be on the preparation of this commercial albumin for horses. Administering it to three horses, performing hematological and clinical analyses, and suggesting the product's safety could be dangerous. Case reports can be reported, but they do not encourage its use.

The authors mention evaluating plasma volume expansion. How was this evaluated? I did not see any assessment of plasma volume expansion in the text that could be considered an appropriate methodology for this purpose.

Furthermore, the introduction is long and contains basic concepts. It needs to be more focused on horses. There is a lot of human data, and there are already studies on horses and albumin administration (lines 50-57; 44-46). Line 77 is missing the reference date. Lines 214-217 need to be written better. Line 247: non-invasive blood pressure.

If the animals' response to the administered product is to be evaluated, a phase 1 and phase 2 clinical trial would need to be conducted, following the steps and an adequate number of animals.

Therefore, the text needs to be adapted to the albumin production and administration reports, and then reevaluated.

Reviewer #2: Dear authors, thank you for submitting a detailed report of a very well constructed and undertaken study.

The manuscript is overall well written with very few typological error and good grammar. However it is percievable that it is not written by an english native. i would suggest an english proof reading as some of the sentences have a litteral translation from the latine way of phrase construction and some of the word choicescould be improved. that would help with the overall comprehension of the manuscript for the reader.

Concerning the content of the manuscript, the M&M is well written and the methodology very thorough.

The results are well presented eventhough there are some discussion elements in this part that should be transfered to the dedicated section. With three horses only (which is a good start), it is difficult to interpret the overall blood parameters variations overtime. You have on one of the three horses blood parameters that seem to behave differently. I would be cautious about overinterpreting these results and a clear and detailled description would be more informative.

It should be stated that you did not document any clinical sign of immediate immune reaction to the albumin infusion. Didi you look for delayed reaction?

In the discussion, the limits of the study are not well described.

I would also encourage you to deepen the discussion using data published in other species like dogs.

Finally, I had a hard time understanding how you evaluated the plasma expansion following the infusion.

Apsect to consider that can interfere with the variation of the blood parameters are : did some of the horses got thirsty during the infudion, this is something we certainly see with the administration of hypertonic saline and if a horse decided to drink more than another, it would have an effect on the Ht and oncotic pressure / plasma volume. Consider also a potential splenic contraction that could explain a short lived increased hematocrit.

Concerning platelet counts and variations, did you use only the automat results or did you also check a blood smear?

Thanks again for the quality of the study, I'm looking forward reading your revisions

**Do you want your identity to be public for this peer review?** For information about this choice, including consent withdrawal, please see our Privacy Policy

Reviewer #1: No

Reviewer #2: **Yes:** Gwenola Touzot-Jourde

---

## [Author Response · Author response to Decision Letter 1]

6 Dec 2025

Dear reviewer, we strongly appreciate all your comments which we are absolutely sure can improve the understanding and quality if our results which are focused in the obtention and preliminary evaluation of the clinical response after intravenous administration of equine albumin in adult Colombian Creole Horses.

1. The submitted data contains all raw data required to replicate the results of your study.

2. The requested figure is attached.

3. The ethical statement is described only in the material and methos section.

Reviewer 1

1. The study proposes the preparation and evaluation of intravenously administered albumin in horses.

2. Title: Does not reflect the study performed.

a. R/ Following the observations, we updated the manuscript title seeking to reflect the aim of the study. See lines 2 – 4.

3. The study objectives do not reflect the study performed.

a. R/ Following the observations, we updated the manuscript aim to address our study results. See lines 80 – 84.

4. The study focuses more on the production of albumin for commercial use. Its use in horses was tested on only three horses, which does not allow for any statistically significant conclusions.

a. R/ It this absolutely important to considered that there is not any available local production or commercialization of equine albumin for veterinary use in Colombia. This is the reason why we also focused our results on the pharmaceutical description and further pharmaceutical evaluation of the product, showing to our community the local capacity (in Colombia) to obtain a high-quality standard final albumin product/formulation and the subsequent preliminary administration in adult horses.

5. The focus of the study, based on the title, should be on the preparation of this commercial albumin for horses. Administering it to three horses, performing hematological and clinical analyses, and suggesting the product's safety could be dangerous. Case reports can be reported, but they do not encourage its use.

a. R/ Following the observations, we avoided the albumin formulation safe recommendation in the abstract, aim and results of the manuscript. Now, according to the international pharmaceutical standard agencies, critical quality parameters (CQA) are directly related to final safe use of pharmaceutical products, and we went overall this premise in our study. We followed International Conference on Harmonization of Technical Requirements for Registration of Pharmaceuticals for Human Use (ICH) and the World Health Organization Standards to Manufacture Medicaments (WHO/MM), identifying patient safety as the main criteria. Regarding the reduce number of individuals used is supported by the unavailable data in Colombia about equine albumin obtentions and administration into horses. These is why this study is aimed to analyze the clinical response in a reduced group of horses before using larger individuals groups. We need to be ethically responsible.

6. The authors mention evaluating plasma volume expansion. How was this evaluated? I did not see any assessment of plasma volume expansion in the text that could be considered an appropriate methodology for this purpose.

a. R/ Although plasma volume expansion was not directly evaluated, the obtained results allowed us to determine the variation in the plasma volume expansion according to Quispe-Cornejo A, et al. and Van Beaumont et al. See lines 469-471.

7. Furthermore, the introduction is long and contains basic concepts. It needs to be more focused on horses. There is a lot of human data, and there are already studies on horses and albumin administration (lines 50-57; 44-46). Line 77 is missing the reference date.

a. R/ Following the observations, we reorganized the introduction and the reference was added. See line 68.

8. Lines 214-217 need to be written better. Line 247: non-invasive blood pressure.

a. R/ Corrected as suggested. See lines 205 – 209.

9. If the animals' response to the administered product is to be evaluated, a phase 1 and phase 2 clinical trial would need to be conducted, following the steps and an adequate number of animals.

a. R/ Our results will support further studies including higher number of individuals to evaluate a full clinical phase I trial and a further phase clinical trial. Following ethical committees’ recommendations in our country and due to the lack of previous information in Colombian Creole Horses, we initiated our studies in small populations.

10. Therefore, the text needs to be adapted to the albumin production and administration reports and then reevaluated.

Reviewer 2

1. Dear authors, thank you for submitting a detailed report of a very well constructed and undertaken study.

a. R/ Ok.

2. The manuscript is overall well written with very few typological error and good grammar. However it is perceivable that it is not written by an english native. i would suggest an english proof reading as some of the sentences have a litteral translation from the latine way of phrase construction and some of the word choicescould be improved. that would help with the overall comprehension of the manuscript for the reader.

a. R/ Ok.

3. Concerning the content of the manuscript, the M&M is well written and the methodology very thorough.

a. R/ Ok.

4. The results are well presented eventhough there are some discussion elements in this part that should be transfered to the dedicated section. With three horses only (which is a good start), it is difficult to interpret the overall blood parameters variations overtime. You have on one of the three horses blood parameters that seem to behave differently. I would be cautious about overinterpreting these results and a clear and detailled description would be more informative.

a. R/ Following ethical committees’ recommendations in our country and due to the lack of previous information in Colombian Creole Horses, we initiated our studies in small populations.

5. It should be stated that you did not document any clinical sign of immediate immune reaction to the albumin infusion. Didi you look for delayed reaction?

a. R/ Yes, we did record delayed reactions 24h, 48h and 72h after the albumin administration. At the moment that the horses went back to the herd, they were observed constantly by our veterinary team (including the horse keeper) during the day for three continuous days. See lines 207 – 209.

6. In the discussion, the limits of the study are not well described.

a. R/ corrected as suggested. See lines 525 – 528.

7. I would also encourage you to deepen the discussion using data published in other species like dogs.

a. R/ We are aware about the different studies available in dogs. But we considered that dogs results should not be comparable with horse results since dogs may be receiving exogenous albumin from a different species, while horse are receiving the albumin from the same species. This may lead, in the case of dogs, to a higher prevalence of adverse reactions. In any case, we added a revision study regarding the use of albumin in dogs. See lines 483-492.

8. Finally, I had a hard time understanding how you evaluated the plasma expansion following the infusion.

a. R/ Although plasma volume expansion was not directly evaluated, the obtained results allowed us to determine the variation in the plasma volume expansion according to Quispe-Cornejo A, et al. See lines 469-471.

9. Apsect to consider that can interfere with the variation of the blood parameters are : did some of the horses got thirsty during the infudion, this is something we certainly see with the administration of hypertonic saline and if a horse decided to drink more than another, it would have an effect on the Ht and oncotic pressure / plasma volume. Consider also a potential splenic contraction that could explain a short lived increased hematocrit.

a. R/ considered. According our methodology and in order to to simulate the usual environmental and behavioral conditions for the animal, after the albumin administration, all three individuals were taken back with the herd with food and water ad libitum.

10. Concerning platelet counts and variations, did you use only the automat results or did you also check a blood smear?

a. R/ We used the results from the laboratory.

11. Thanks again for the quality of the study, I'm looking forward reading your revisions.

a. R/ Thank you for your helpful and inspiring comments.

---

## [Editor Report · Decision Letter 1]

8 Jan 2026

Obtention and preliminary clinical evaluation of an equine albumin for intravenous administration in adult Colombian Creole Horses (Equus ferus caballus).

PONE-D-25-40618R1

Dear Dr. S.Estrada-Gomez,

We’re pleased to inform you that your manuscript has been judged scientifically suitable for publication and will be formally accepted for publication once it meets all outstanding technical requirements.

Kind regards,

Claudia Interlandi, Ph.D

Academic Editor

PLOS One

Additional Editor Comments (optional):

The paper has been improved and can be accepted for publication.
---

## [Editor Report · Acceptance letter]

PONE-D-25-40618R1

PLOS One

Dear Dr. Estrada-Gomez,

I'm pleased to inform you that your manuscript has been deemed suitable for publication in PLOS One. Congratulations! Your manuscript is now being handed over to our production team.

Kind regards,

on behalf of

Professor Claudia Interlandi

Academic Editor

PLOS One